# Could Immersive Virtual Reality Facilitate the Recovery of Survivors of Critical Illness? A Systematic Review

**DOI:** 10.3390/healthcare13222942

**Published:** 2025-11-17

**Authors:** Irini Patsaki, Dimitra Tzoumi, Marios Kalyviotis, Akylina Despoti, Eleftherios Karatzanos, Serafim Nanas, Eleni Magira

**Affiliations:** 1Department of Physiotherapy, University of West Attica, 12243 Egaleo, Greece; 21st Critical Care Department, General Hospital of Athens “Evaggelismos”, 10676 Athens, Greece; 3Laboratory of Clinical Ergospirometry, Exercise and Rehabilitation, Medical School, National and Kapodistrian University of Athens, 11527 Athens, Greece; 4National Rehabilitation Center, 13122 Athens, Greece

**Keywords:** critical illness survivors, post-ICU, virtual reality, post-intensive care syndrome, cognitive rehabilitation

## Abstract

**Background/Objective**: Post-intensive care syndrome (PICS) is a multifactorial, multidimensional condition common among patients who survive critical illness with protracted intensive care unit (ICU) length of stay. Survivors often present physical, cognitive, and psychological impairments that can arise during ICU hospitalization. Virtual reality has emerged as a promising tool in the healthcare field, as it offers innovative solutions to the challenges faced by critically ill survivors. We think that VR might help support people recovering from PICS at home, and this study aims to explore the benefits across the spectrum of PICS to describe the technological characteristics that could support and augment these interventions and present clinical recommendations. **Methods**: This systematic review searched PubMed, the Cochrane library, Science Direct, and Scopus databases up to July 2025. In this study we included randomized controlled trials (RCTs) examining the impact of VR on PICS. The methodological quality was assessed with the PEDro scale for RCTs and with NOC for non-RCTs. **Results**: A total of five studies met the inclusion criteria and were included. Three were RCTs, one non-RCT, and one series of cases. The studies presented good methodological quality. Virtual reality was found to be safe for critically ill survivors. All aspects of PICS were examined, with positive results in recovery of psychological disorders, such as anxiety and PTSD, and muscle strength as assessed by hand grip and cognition. The main limitation could be the limited number of RCT studies due to the novelty of the intervention. **Conclusions**: Virtual reality technology could be safely implemented in the field of post-ICU recovery and effectively assist the rehabilitation of physical, cognitive, and mental disorders of ICU patients. The protocol was registered in the Open Science Framework registry.

## 1. Introduction

Surviving critical illness often involves a long and challenging recovery. Although rehabilitation starts from within the intensive care unit (ICU), the multidimensional nature of its complications requires a multidisciplinary approach. The term post-intensive care syndrome (PICS) describes the impairments that the survivors could manifest, which could be physical, cognitive, or psychological. These could be presented alone or in combination [1]. The occurrence rate of these post-ICU sequelae reaches 70% of survivors [2]. These have a tremendous impact on functional ability and quality of life, often leading to challenges in regaining previous levels of independence and psychosocial functioning [3]. In either case there is a need for a rehabilitation strategy tailored to the needs and abilities of each patient. A thorough assessment, even before discharge from intensive care, will determine the degree of assistance and rehabilitation that will be required to carefully plan the recovery, starting from selecting the most appropriate facility [3]. Over the past decades, several rehabilitation strategies have been implemented in ICU survivors to investigate potential beneficial effects in their recovery. These rehabilitation programs involved exercises or neuromuscular stimulation, combined with nutrition or psychological support [4]. Recent guidelines have underlined the importance of multimodal rehabilitative strategies, with four strong recommendations across all three domains of impairment. Additionally, four treatment options were proposed for physical impairments [5]. The need for a wide range of follow-up service strategies for critically ill survivors to improve their long-term outcomes is most significant. Yet, post-ICU follow up programs face various obstacles, either due to the lack of resources or time and distance challenges of the survivors and their caregivers. These barriers, along with suggested facilitators, have been well documented [6,7]. Advances in technology have offered new and innovative means of rehabilitation. Telerehabilitation has been proposed as a solution even in chronic respiratory diseases, presenting the same efficiency as traditional pulmonary rehabilitation programs [8]. Virtual reality (VR) has augmented the field of telerehabilitation through technological features that increase engagement and provide a more dynamic experience. Virtual reality rehabilitation programs have been implemented within the ICU in an effort to reduce the incidence of stress, anxiety, delirium, and physical decline [9]. Thus, VR has been explored as a means to prevent PICS. Additionally, there is a growing interest in exploring its effect to assist recovery following ICU discharge. VR could be a solution to facilitate home-based programs that could address both the psychological and physical challenges of PICS [10]. This could further advance the provision of telerehabilitative services, overcoming barriers related to physical distance. This study aimed to systematically review and explore the benefits across the spectrum of PICS to describe the technological characteristics that could support and augment these interventions and present clinical recommendations.

## 2. Methods and Materials

The present study was conducted following the Preferred Reporting Items for Systematic Review and Meta-Analyses (PRISMA) guidelines [11].

The protocol was registered in the Open Science Framework registry (DOI:10.17605/OSF.IO/4K53C).

### 2.1. Search and Study Identification

A thorough search for appropriate published studies was performed by two authors (K.M. and I.P.) independently in PubMed, the Cochrane library, Science Direct, and Scopus up to July 2025. The search terms used followed the PICO framework, and were as follows: critical illness survivors, post-ICU patients, critical ill patients, virtual reality, augmented reality, exergames, serious games, post-intensive care syndrome.

All the terms were used in different combinations to create search strategies that were applied in the selected databases (Table 1).

Additionally, a hand search was performed across all reference lists of the identified articles.

The inclusion criteria for studies in this systematic review were defined as follows: (a) clinical trials or RCT or non-RCT; (b) participants of 18 years of age or older; (c) survivors of critical illness; (d) in all studies the intervention group must have followed a fully immersive VR training program; (e) outcomes related to PICS such as muscle strength, functional ability, anxiety, depression, post-traumatic stress disorder (PTSD), cognitive function, or quality of life; (f) be written in English.

Exclusion criteria from the research study were: (a) VR programs implemented in the ICU; (b) narrative or systematic reviews, study protocols, posters, abstracts, and single case studies as they cannot be studied systematically; (c) other pulmonary diseases; (d) studies where the intervention was designed for training purposes; (e) not fully available text.

A thorough review of the titles and abstracts of studies published in the databases used was performed by two authors (K.M. and I.P.) independently. For those studies that met the criteria according to title and abstract, a further assessment was performed on the full text. Additionally, the reference lists of pertinent literature were examined for potentially relevant articles published in English. Any discrepancies were discussed and resolved by consensus between the two reviewers or by a third (D.T.) when needed.

### 2.2. Methodological Quality

The methodological quality of the included studies was assessed using the PEDro (Physiotherapy Evidence Database) scale, a valid and reliable tool [12,13]. The scale consists of 11 items, 10 of which contribute to the total score. Each item is rated as either “yes” (1 point) or “no” (0 points). The first item relates to external validity and is not included in the final PEDro score. Studies were classified as high quality (≥7), moderate quality (4–6), or low quality (≤3) based on the total score.

The methodological quality of non-randomized studies was assessed using the Newcastle–Ottawa scale (NOS) [14], a validated tool for evaluating the risk of bias in such studies. The NOS consists of eight items grouped into three domains: selection, comparability, and outcome. Each item includes a set of response options, and a star system is applied to provide a semi-quantitative evaluation of study quality. One star can be awarded for each item, except for the comparability domain which allows up to two stars, yielding a total possible score ranging from 0 to 9 stars [14]. The NOS was converted to the Agency for Healthcare Research and Quality (AHRQ) standards, classifying studies as good, fair, or poor. Good quality requires 3–4 stars for selection, 1–2 stars for comparability, and 2–3 stars for outcome. Fair quality needs 2 stars for selection, 1–2 stars for comparability, and 2–3 stars for outcome. Poor quality requires 0–1 star for selection, or 0 stars for comparability, or 0–1 star for outcome. This grading approach enabled a consistent and transparent evaluation of the methodological rigor of the included non-randomized study.

The methodological quality of the case series study [15] was assessed using the Joanna Briggs Institute (JBI) critical appraisal tool. This tool comprises ten questions that evaluate internal validity and the risk of bias in case series designs, with particular attention to confounding, selection bias, information bias, and the clarity of reporting [15].

### 2.3. Data Collection

Clinical information regarding the patients’ demographics (age and sex) and ICU stay (diagnosis upon ICU admission, duration of MV, and duration of ICU stay) was extracted from the studies. All results related to post-intensive care syndrome like muscle strength, functional ability, mental or cognitive disorders, psychological disorders (anxiety, depression, and post-traumatic stress disorder), and quality of life after the completion of the interventions were noted to assess the effectiveness of implementing virtual reality in post-ICU patients. Data on patients’ safety were also examined.

## 3. Results

### 3.1. Identification of Studies

The initial search of online databases identified 234 articles. After removing duplicates, 189 studies remained and were assessed for inclusion. Only five studies [16,17,18,19,20] were considered eligible and were included in the review. From these, one was non-RCT, three were RCTs, and one was a series of cases. The specific study selection flow chart is presented in Figure 1.

### 3.2. Methodological Quality

Three randomized control trials [17,18,19] were assessed according to the PEDro scale and presented a mean score of 5 and 7 (Table 2). Two studies presented a good methodological quality and one a fair. In general, we noted a moderate methodological quality. A major concern could be identified in the blinding of subjects, assessors, and therapists, which was absent in all studies. Additionally, the absence of intention-to-treat analysis could be regarded as a risk of bias, as its application provides a more accurate picture of the interventions’ effectiveness.

According to the Newcastle–Ottawa scale, the study of Despoti et al. [20] was rated with a good methodological quality with an overall of 7 stars (3 for selection, 2 for comparability, and 2 for exposure).

According to the JBI critical appraisal tool, the study by de Vries et al. [16] scored positive on eight questions. The information provided by the authors was unclear in answering the following: (a) the inclusion of consecutive participants, and (b) the inclusion of all participants.

### 3.3. Description of Studies

#### 3.3.1. Population

A total of 315 survivors of critical illness participated either after ICU or following hospital discharge. The patients were divided into COVID-19 patients (*n* = 89) admitted into the ICU and a mixed ICU (*n* = 226) population with a wide range of diagnoses, including sepsis, respiratory, cardiovascular, neurological disorders, or being admitted post-surgery. Additionally, their comorbidities included COPD, hypertension, diabetes, etc. All had a mean duration of ICU length of stay of up to 15 days, with the duration of intubation ranging from 4 to 10 days. The age of the participants ranged from 50 to 85 years, with most of them being elderly. This explains the use of this technology to primarily target cognitive deficiencies. The patients were included a few days after ICU discharge [18,19], following hospital discharge [20], and up to a few months post-hospital discharge [15]. Two studies [17,20] included participants with PICS. All information is presented in Table 3.

#### 3.3.2. Interventions

All studies included in this systematic review utilized a fully immersive system of virtual reality and most of them were targeted cognitive or mental rehabilitation.

Two studies [17,18] utilized a 14 min informational video in which the patient is exposed to an ICU environment and receives voice-over explanations regarding various aspects of the surrounding ICU environment and treatment. The control group utilized the same time relaxing by watching a natural environment. Dongs et al. [19] used a digital operating system to provide motion control, cognition, and activities of daily living training. This system incorporated the latest technology on touch multiscreen, virtual reality, and human–machine interaction. This system targets memory, attention, visual space, calculation, and hand–eye coordination through the completion of virtual tasks.

The intervention group also participated in conventional sessions, like music therapy, aerobic training, and mental health interventions targeting psychological problems. These were also provided to the control group. Despoti et al. [20] introduced two virtual scenarios (animals on a farm and shape–color) of cognitive training to the experimental group, which were first explored in a stroke population [21]. In contrast, the control group followed a conventional program, with pencil and paper, such as word finding, crosswords, or semantic fluency tasks (like naming as many animals as possible in one minute), and drawing shapes and puzzles. Both scenarios practice attention, concentration, information processing, speed processing, visual memory, and visual–spatial functions along upper limb movement and neuromuscular coordination. In de Vries et al. [16] the intervention aimed to augment upper limb functionality by incorporating a puzzle. To train limb movements into a virtual environment, the VR headset had four outward-facing cameras that tracked hand movements using inside–out tracking and computer vision techniques.

Despoti et al. [20] and de Vries et al. [16] developed their games through sessions that involved stakeholders. They both incorporated a multiprofessional team to identify areas that could benefit from the implementation of these systems such as cognitive function and functional ability [16,20]. Additionally, Vlake et al. [17,18] designed a scenario or a film script that introduced the patient to the ICU environment, explaining all devices and noises in an ICU, explaining the need for intubation, mechanical ventilation, central/peripheral lines, gastric tubes, and the various interventions and treatments necessary for overcoming the incidence of critical illness. This facilitates the provision of information that a critical ill survivor needs to understand what happened to him/her within the ICU. On the other hand, the control group could enjoy more relaxing scenery. Thus, the aforementioned studies have used custom-made software. All the included studies used a head-mounted display (HMD) to achieve patient immersion in the virtual environment. The interface with the virtual objects was made feasible as hand movements were tracked using inside–out tracking and computer vision techniques, using cameras on the front of the VR headset [16,19,20].

#### 3.3.3. Feasibility and Safety

Although the feasibility of using virtual reality in critically ill patients has been examined within the ICU [22,23], three of the included studies examined it in survivors as well [16,18,20]. None of them reported adverse events. More specifically, Vlake et al. [18] examined the occurrence of cybersickness (simulator sickness questionnaire), adverse events, and changes in vital signs. The authors noted that none of the participants presented severe symptoms of cybersickness and that there were no changes in their vital signs. VR sessions were not interrupted or discontinued due to side effects. The cybersickness scores were low in both groups, and none of the symptoms were still present 15 min after taking off the VR headset and exiting the virtual environment. Despoti et al. [20] used the suitability evaluation questionnaire to examine the safety, usability, and negative effects that may occur during the use of VR, but did not report any serious incidence. In general, virtual reality could be safely implemented in this population during their hospitalization, follow-up clinic, or even at a rehabilitation facility.

#### 3.3.4. Effects of Intervention

Cognitive function was examined in three controlled studies that used different assessment tools like the mental component of SF-36, MOCA, MMSE, FAB, and ACE-r. Specifically, ACER-r assesses several different domains of cognitive function like attention, memory, fluency, language, and visuospatial abilities. All studies showed a significant improvement in their population, with three of them reaching a statistically significant difference between groups in favor of the experimental intervention. Although ACE-r did not reach a significant difference between groups, the authors noted that certain domains, like fluency and visuospatial abilities, were achieved.

Regarding psychological sequelae, the studies examined depression [17,18,20], anxiety [17], and post-traumatic stress disorder (PTSD) [17,18]. In Vlake et al. (2022), although at the 3-month follow-up the control group reported significantly higher probable anxiety (*p* = 0.02), the HADS score did not show this [17]. Significant differences between groups regarding depression were noted only in one study [18]. The authors reported that depression scores decreased over time for all patients, with the experimental group resulting in lower depression scores than the control one throughout the whole study period [18]. Similar findings were observed for PTSD [18]. Additionally, Vlake et al. (2022) reported only a general improvement at 6 months after hospital discharge, without this being attributed to randomization [17].

Physical and functional recovery was examined in two studies [16,18]. The series of cases showed no difference in hand grip strength after the completion of the intervention (4 weeks), as measured in the Morton mobility index [16]. The physical component of SF-36 showed significant differences only at 4 months of follow-up [18].

Health-related quality of life was examined in three studies [17,18,19]. One study reported a significant difference at 3 months of follow-up, with this remaining during the 4 and 6 months of follow-up [17]. Dong et al. (2023) noted a statistically significant improvement in the experimental group compared to the control group across all domains of SF-36, except for the domain assessing pain [19].

## 4. Discussion

The prevention and rehabilitation of PICS depend on a multidisciplinary approach involving various healthcare professionals [24]. Technological innovation has a lot to offer, and some studies have already investigated the preventive effectiveness of VR technology by implementing this kind of intervention in ICUs [25].

To our knowledge, this is the first study to examine the overall effect of immersive VR in survivors of critical illness and, specifically, as a therapeutic tool for PICS. Positive findings were mostly noted in the domains of cognitive and psychological sequelae, like depression, anxiety, PTSD, and cognitive function. Physical impairments were not extensively examined. Yet, we were able to note a positive effect in functionality, but not in muscle strength. Considering the novelty of this therapeutic intervention, the limited number of studies included is well understood. The small number of included studies and the heterogeneity of the outcomes presented do not allow for the generalization of these preliminary findings; however, the various domains that may benefit from this therapeutic intervention are presented. Additionally, most studies examined safety, feasibility, and the occurrence of side effects, without manifesting anything alarming.

The virtual environment offers versatile and multisensory effects in regard to combined cognitive and physical exercises, under a safe environment, including an entertaining aspect. Fully immersive VR in particular uses multisensory features [26] to present specific and targeted tasks that provide cognitive training even such impairments [27]. The participants have to follow specific orders to successfully complete their task, with a variant degree in the scenario difficulty to continuously offer the desired training effect. The results were found to improve executive function, memory, and attention. The dynamic interface between different sensory inputs, motor responses, and cognitive involvement triggers a cascade of neuroplastic changes, altering or enhancing synaptic connections, thereby increasing neuroplasticity [28,29]. These promising results are well documented in healthy older adults with mild cognitive impairment and various chronic pathologies that also include cognitive decline in their symptoms like Parkinson’s disease [30] or stroke [31]. Despoti et al. [20] integrated a novelty into their design by introducing a virtual non-player character (NPC), who explains the tasks and instructs and guides the patient throughout the game. By repeating and explaining the tasks, the NPC provides assistance and reduces interferences outside the virtual environment.

Regarding the psychological impact, the use of an informative scenario as presented by Vlake et al. [17,18] allows the patient to complete the “puzzle” of their ICU stay. This has been successfully introduced before via ICU diaries. As the patient starts to have a clear picture of their stay in the ICU environment, a significant decrease in post-traumatic stress disorder, anxiety, and depression has been noticed [32,33]. Also, ICU survivors may recall stressful memories from the ICU, such as factual, emotional, and delusional memories [34]. ICU diaries and virtual reality informative scenarios summarize what happened in the ICU. Thus, any factual memories are replaced by real information. This helps patients understand the sequence of the real events that may have occurred during their stay in the ICU. Additionally, participation in an enjoyable task alleviates anxiety and depressive feelings. Every patient has their own pathway of recovery in an ICU. What needs to be emphasized is that virtual reality provides more realistic imagery, offering a detailed presentation of medical procedures, explanations of ICU equipment and noises, as well as introductions to the roles of ICU personnel.

Although the physical consequences of surviving critical illnesses, like muscle disuse and weakness, were not widely studied, it is important to emphasize that the interaction in the virtual environment could be passive, assisted, or fully active. Thus, the effort that needs to be made could change in relation to the patient’s status, depending on the progress made. This approach allows for customizing the training according to the specific needs of each patient during each session. It has also been observed that the use of robotic systems assisted the mobilization of the affected limb [35]. Additionally, during an ICU stay we have seen the combined utilization of cycling ergometry with VR. This allows control of the effort needed by the patients to perform the desired task [36]. From a physiological aspect, an MRI study showed significant gray matter increase in five brain areas: the tail of the hippocampus, the left caudate nucleus, the rostral cingulate zone, the depth of the central sulcus, and the visual cortex [37]. It was also found that the gray matter volumes of motor, premotor, and supplementary motor cortices correlated positively with the power and active range of motion measured in motor tests [37,38]. Different cortical adaptations in EEG activities strongly indicate underlying neural plasticity and neural reorganization [39]. This may explain the positive results that have been noted in the physical performance of the patients.

A recent systematic review that examined the effect of virtual and augmented reality in stress and anxiety reported a modest reduction, possibly due to increased heterogeneity [9]. The authors stated high participation among patients, which suggests that this kind of intervention is feasible and well-tolerated [9]. Even mediational virtual reality has been used in order to alleviate the stress of hospitalization in an ICU. The authors used “RelaxVR” to provide patients with a calm immersive scene, like rolling waves on a beach, with voice-guided meditation that promoted breathing control and relaxation [40]. Additional positive results have been noticed in the domain of motor-functional recovery. A systematic review and meta-analysis of 11 randomized control trials showed that VR compared to more conventional rehabilitation techniques improved the patients’ overall motor and functional ability [41]. Ten years ago, the initial idea of implementing innovative technologies of rehabilitation in ICUs was presented through video games like Nintendo Wii and Wii fit that were used to train balance, strength, and endurance [42].

In conclusion, virtual reality rehabilitation has been presented as a safe environment for both cognitive and physical rehabilitation [43] and in more frail populations like the elderly [44]. These technologies can simulate real-world scenarios in a controlled environment, allowing all participants to interact and practice safely. Especially for tasks and scenarios where the patient is not seated when they are training balance or gait, the use of harnesses is mandatory. The path to recovery is often long, and virtual environments play a valuable role in keeping patients engaged and focused throughout the process. By offering realistic representations of activities of daily living, virtual reality brings a new perspective to functional recovery. The entertaining elements embedded in these scenarios are a key factor in enhancing patient engagement and motivation, ultimately encouraging greater effort during rehabilitation.

### Clinical Considerations

Virtual reality presents promising results as both a preventive and rehabilitative intervention for post-intensive care syndrome. Feasibility is well established; however, research is needed to fully investigate effectiveness across the entire spectrum of PICS. Patients noted an increased level of satisfaction, which indicates their ability to interact successfully in the virtual environment [16]. This is mostly attributed to newer software technology that allows more accurate motion detection and interaction. This is an important feature when considering elderly populations who are not familiar with technology. What we need to consider further is the training environment of the critically ill patients; we understand the need for custom-made scenarios and games, tailored to the needs of this specific population. From a clinical perspective, there are various components of the virtual environment that should be carefully considered and properly designed. This could have a significant limitation regarding the cost of a custom-made game, which leads to a significant access restriction. Conversely, commercial games are easily accessible and have quite a low cost.

User-centered Environment Design: The most significant part of the process of developing a virtual reality game is the environment itself. Gerber et al. have examined the effect of different environments, with the natural environment having the highest positive and restorative effect on the physiological and psychological state of the participants. As the ICU is a noisy and stressful environment that often causes sensory overload, it is easily understood that a more neutral scenery is needed [45]. Yet, the game could have various scenarios, from plain to more elaborate, as part of a classification in relation to the patients’ progress so as to continuously attract his/her interest or create an anticipation for the next session. Moreover, the design should be associated with the goal of the VR session. When the goal is to distract patients from a painful stimulus or environment, then a more elaborate scenery could be more effective; when the performance of a guided task is the goal, a simpler environment could be beneficial [46]. Interacting in a virtual environment is not always easy, especially for the older population; therefore, providing a simple task as part of training sessions should be explored. Of the studies reviewed, Vries et al. [16] implemented a VR environment which simulates a relaxing living room and gives you an idea of being at home. But one could also think that all these different colors and lighting could distract patients’ attention. On the other hand, Despoti et al. [20] presented one simple (shape–color) and a more elaborate one (farm). Notably, both prototype scenarios were designed to fit the goals of the study.

Interface Design: The medium of communication between people and machines to support optimal interaction has advanced over the past few years. New headsets offer both visual and auditory information. These are also equipped with tracking devices that could recognize the position of arms in space, thus minimizing the use of external devices. Real-time interaction and a fully immersive experience are key factors for augmenting our patients’ experiences [47]. Simplifying our interfaces could assist critically ill patients whose consciousness level often varies, and of course specifically the elderly population who are increasingly present in ICUs. Artificial intelligence and machine learning have played a vital role in advancing human–computer interaction, ensuring safety and increasing patients’ trust [48,49].

Levels of difficulty: Apart from offering a relaxed or informative scenario, there is also a training character that should not be overlooked. Thus, for the game to provide the appropriate stimulus that is needed for training purposes, it should include tasks with varying levels of difficulty [50,51]. Over the past few years, significant efforts have been made to integrate artificial intelligence (AI) into rehabilitation systems in order to offer a more personalized treatment. Specifically, explainable AI represents a critical advancement in the application of AI for data-driven clinical decision support. AI could constantly adapt the level of difficulty and interaction in accordance with data gathered from each patient and during each session [48,49]. Additionally, an AI-based virtual assistant could effectively guide the patients as the level of difficulty changes and constantly provide feedback in order to increase interaction and engagement [48,49].

## 5. Limitations

At this point, only a limited number of randomized controlled trials are available concerning post-ICU rehabilitation. The sample size of the included population is often limited and not representative of the ICU population. The protocols that were implemented and the outcome measures that were assessed are very heterogeneous. Physical functioning and activities of daily living were not fully explored, alongside other symptoms related to post-intensive care syndrome, like pain, sleep disorders, endurance, etc. Thus, definitive conclusions cannot be drawn and these results cannot be generalized.

In general, the limited number of studies that were included in this systematic review could be attributed to various reasons. First, it could be due to the emerging nature of fully immersive virtual reality interventions for survivors of critical illness. Despite employing a comprehensive and systematic search across multiple databases, only a few studies met the inclusion criteria that specifically addressed immersive VR in this population. Secondly, our aim to include mostly high-level evidence and not conference papers, as they present limited data, could also have influenced this result. Additionally, we note that language restrictions may have limited the number of identified studies, especially from countries that have shown significant technological advancement, such as China and Japan. VR interventions for survivors of critical illness are a promising field that should be explored further.

## 6. Conclusions

Virtual reality technology could safely be implemented in the field of post-ICU recovery. It could be implemented for both educational and training purposes, providing a more enjoyable means of rehabilitation. Promising results were identified in enhancing the rehabilitation of physical, cognitive, and mental disorders in critically ill survivors.

The ability to provide dual task training and to combine both cognitive and physical components seems to augment training stimulus. Further research is needed to establish effectiveness and fully explore the sequelae of post-intensive care syndrome. Little evidence has been provided regarding functional rehabilitation and especially activities of daily living. Additionally, the ICU population is quite heterogeneous, thus we need RCT studies with bigger populations that could fully describe all pathologies. As technology evolves, further investigation of designing features, software, and ways of enhancing interaction is of high importance, especially AI and decision-supporting systems. Home-based training and monitoring of this population could also increase accessibility to such services.

## Figures and Tables

**Figure 1 healthcare-13-02942-f001:**
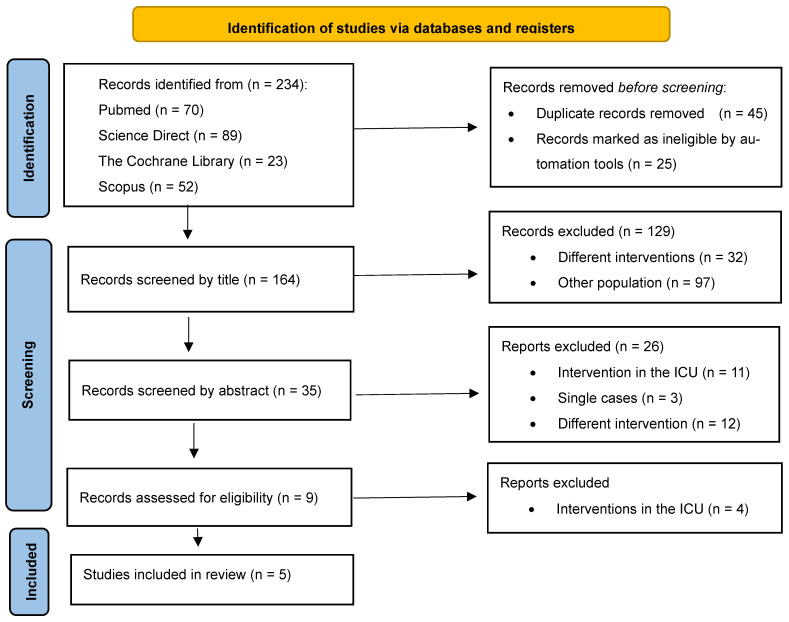
Prisma Flow diagram of the included studies.

**Table 1 healthcare-13-02942-t001:** Search strategies applied in PubMed.

**Search strategy**	[(critical illness survivors) OR (post-ICU patients) OR (post-intensive care unit patients)] (critical ill patients) OR (ICU) OR (critical illness) [(virtual reality) OR (VR) OR (augmented reality)] [(exergames) OR (serious games)] [(Post-intensive care syndrome) OR PICS]
**Search string**	# 1 OR # 2 AND # 3 AND # 5 # 1 OR # 2 AND # 4 AND # 5

**Table 2 healthcare-13-02942-t002:** PEDro assessment of included studies.

	1 *	2	3	4	5	6	7	8	9	10	11	
Vlake et al., 2021 [18]	Y	Y	Y	Y	Ν	Ν	Ν	Y	Ν	Y	Y	6/10
Vlake et al., 2022 [17]	Y	Y	Y	Y	N	N	N	Y	N	Y	Y	6/10
Dong et al., 2022 [19]	Y	Y	N	Y	N	N	N	Y	N	Y	Y	5/10

* The eligibility criteria item does not contribute to the total score. (Y: YES, N: NO).

**Table 3 healthcare-13-02942-t003:** Characteristics of the studies included in the systematic review.

Study(Design)—Country	Population	Intervention	Control	Outcomes	Results (Between-Group Comparison When Applicable)
De Vries et al., 2025 [16] (pre-post study)—Netherlands	10 patients from ICU to hospital discharge	4 VR therapy sessions (puzzles) (3 times/week, 20 min)	-	Morton Mobility index Hand Grip	MMI: *p* < 0.05 HG: *p* = ns
Vlake et al., 2022 [17] (RCT)—Netherlands	COVID-19 3 months post hospital discharge (IG: 45/CG: 44)	14 min-long informational video	Scenery VR	IES-r HADS SF-36 EQ-5D	Psychological distress: *p* < 0.05 at 3 months but not at 4 and 6 months HADS and IES-r: *p* = ns at all points.
Vlake et al., 2021 [18] (RCT)—Netherlands	(IG: 25/CG: 25) After ICU discharge	Informational video	Scenery	Immersive Tendencies Questionnaire, Simulator Sickness Questionnaire, Presence Questionnaire, IES-r, BDI-II, MCS-12, PCS-12, EQ-5D Adverse events	Only IES-R, BDI and MCS-12 (*p* < 0.05) at 1 month No simulation sickness, no change in vital signs, no adverse events
Dong et al., 2023 [19] (RCT)—China	(IG: 68/CG: 68) After ICU discharge with cognitive impairment	Cognitive rehabilitation + Music Therapy + Aerobic Training	Music + Aerobic training	MOCA SF-36	*p* < 0.05 3 and 6 months
Despoti et al., 2025 [20] (nonRCT)—Greece	(IG: 15/CG: 15) Post Hospital discharge– PICS populations	2 VR games 12 sessions (3 times a week, for 4 weeks). Each session lasted 30 min	Pencil–paper cognitive training	ACE-R (general cognitive function), MMSE (general cognitive function), FAB (executive function), GDS (depression)	Only Visuospatial abilities and fluency *p* < 0.05

IG: intervention group; CG: control group; PICS: post-intensive care syndrome; VR: virtual reality; ICU: intensive care unit, ACE-R: Addenbrooke’s cognitive examination revised; BDI: Beck depression inventory; EQ-5D: European quality of life 5D questionnaire; FAB: frontal assessment battery; GDS: the geriatric depression scale; HG: hand grip; HADS: hospital anxiety depression scale; IES-r: impact event scale-revised; MMSE: mini-mental state examination; MMI: Morton mobility index, MOCA: Montreal cognitive assessment; MCS-12: mental component scale of the short-form 12; PCS-12: physical component scale of the short-form 12; SF-36: 36-item short form health survey scale.

## Data Availability

No new data were created or analyzed in this study.

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
