# Peer review of "Could Immersive Virtual Reality Facilitate the Recovery of Survivors of Critical Illness? A Systematic Review"

_healthcare, 2025, doi:10.3390/healthcare13222942_

Round 1

Reviewer 1 Report

Comments and Suggestions for Authors

The manuscript entitled “Could immersive virtual reality facilitate recovery of survivors of critical illness? A systematic review” was interesting. The authors aimed to explore the benefits of the technology across the spectrum of PICS, to describe the technological characteristics that could support and augment virtual reality interventions, and to present clinical recommendations. However, some issues need further attentions:

  • Please choose the keywords using the MeSH terms.
  • In the introduction section, the gap in the existing knowledge needs to be highlighted and other similar studies should be reviewed.
  • Please ensure that the aim of the study is similar in the abstract and introduction in terms of the wordings.
  • Please mention the inclusion criteria in the abstract.
  • Please mention the tools used for risk of bias assessment.
  • Please add the timeline of the study.
  • The number of the included studies is very limited, and affect any conclusion derived from the study.
  • Figure 1 should be completed based on the PRISMA guideline. In the current form, the diagram has not been filled correctly.
  • A synthesis after reporting the results is required.
  • The conclusion section is so brief. Please revise and expand it to include the study results and recommendations for future research.
  • Please upload the search strategies as a supplementary file.

Author Response

We would like to thank the reviewer for looking into our work and providing cumulative comments.

  • RQ: Please choose the keywords using the MeSH terms.

A: Although the use of MeSH terms provides more precise and comprehensive presentation than simple keywords, there are words to describe our population that aren’t presented as MeSH terms. But in order to  better identify our study we include the following: Critical illness, Quality of life  and cognitive training instead of rehabilitation.

  • RQ: In the introduction section, the gap in the existing knowledge needs to be highlighted and other similar studies should be reviewed.

A: Thank you for bringing this to our attention. We further enriched our introduction to underline the importance of technology in the rehabilitation of critically ill patients. (Introduction lines 62-69) and the progression presented in this field. We also included these recent systematic reviews:

 Cox, N. S., Dal Corso, S., Hansen, H., McDonald, C. F., Hill, C. J., Zanaboni, P., Alison, J. A., O'Halloran, P., Macdonald, H., & Holland, A. E . Telerehabilitation for chronic respiratory disease. Cochrane Database Syst Rev. 2021;1(1):CD013040.

  Savoric, T., Aziz, S., Ling, R. R., Antlej, K., Arnab, S., & Subramaniam, A. Systematic review: The impact of virtual reality interventions on stress and anxiety in intensive care units. J Crit Care. 2025;90:155164.

  • RQ: Please ensure that the aim of the study is similar in the abstract and introduction in terms of the wording.

A: Appropriate changes were made to maintain consistency.

  • RQ: Please mention the inclusion criteria in the abstract.

A: Inclusion criteria are presented in the abstract in short form so as to keep world limits. Quote: “In this study we included randomized controlled trials (RCTs) examining the impact of VR on PICS”

  • PQ: Please mention the tools used for risk of bias assessment.

A: Risk of bias assessment in the form presented in meta-analysis wasn’t performed. Yet, PEDro scale could present related information and has been widely used. We added the following in the results “Blinding was absent in all studies” as this could be noted as significant bias. And JBI offered relevant information that are noted in the manuscript.

  • PQ: Please add the timeline of the study.

A: Although this is mentioned in the abstract, thank you for pointing out that we forgot it in the main text. We added the time specification for included studies “up to July 2025”.

  • PQ: The number of the included studies is very limited, and affect any conclusion derived from the study.

Α: Implementation of VR technology is quite new in this population. Thus our goal was to explore the preliminary findings (which turn out to be promising), as well as the potential of VR as a means of post ICU rehabilitation and PICS syndrome management. Describing the technological designs used is also important to consider for future studies. We do agree that there has been limited data on the topic so far, which affects any conclusion. So we have appropriately included and discussed this point in the discussion

  • RQ: Figure 1 should be completed based on the PRISMA guideline. In the current form, the diagram has not been filled correctly.

Α: We corrected Figure 1 and added information- data missing.

  • RQ: A synthesis after reporting the results is required.

A: As the reviewer has pointed out, the included number of studies is limited and the outcomes are quite heterogeneous. A qualitative synthesis was presented to the extent that this was feasible, as a quantitative wasn’t. 

  • RQ: The conclusion section is so brief. Please revise and expand it to include the study results and recommendations for future research.

A: Conclusions were further enriched and recommendations  for future studies were added. We quote “Virtual reality technology could safely be implemented in the field of post-ICU recovery. It could be implemented for both educational and training purposes, providing a more enjoyable means of rehabilitation. Promising results were identified in enhancing the rehabilitation of physical, cognitive, and mental disorders of critically ill survivors.

The ability to provide dual task training and to combine both cognitive and physical components seems to augment training stimulus. Further research is needed to establish effectiveness and fully explore the sequelae of post-intensive care syndrome. Little evidence has been provided regarding functional rehabilitation and especially activities of daily living. Additionally, ICU population is quite heterogeneous, thus we need RCT studies with bigger populations that could fully describe all pathologies. As technology evolves, further investigation of designing features, software and ways of enhancing interaction is of high importance, especially AI and decision supporting systems. Home-based training and monitoring of this population could also increase accessibility to such services” lines 457-469.

  • RQ: Please upload the search strategies as a supplementary file.

A:We included a search strategy in the manuscript so as to have all information presented and avoid supplementary files for readers' convenience. We added our pubmed search strategy as an example, as this database offers more complex strategies. Yet, if the reviewer feels that further information were needed, we could add a supplementary file.

Reviewer 2 Report

Comments and Suggestions for Authors
  1. This is a systematic review manuscript, thus, it is crucial to comply to PRISMA guideline, details such as the exact search strings, date ranges, and any language restrictions should be included to enhance reproducibility and transparency of the systematic review process.
  2. Include PRISMA checklist.
  3. Three distinct quality assessment tools (PEDro, NOS, and JBI) were included. However, the rationale for choosing different tools for different study designs should be clarified. 
  4. The review includes only five studies with relatively small and heterogeneous samples (different populations, interventions, and outcomes). This limitation should be more explicitly acknowledged in the discussion, emphasizing how heterogeneity affects the generalizability of the conclusions regarding VR’s efficacy in post-ICU rehabilitation.
  5. Explain why meta-analysis is not performed/ included in the manuscript.
  6. The results indicate improvements in psychological, cognitive, and physical outcomes, however, the discussion does not fully differentiate the strength of evidence among these domains.
  7. The manuscript does not discuss the potential for publication bias or language bias, which could influence the synthesis since only English-language studies were included. Further explain/ justify this item.
  8. Conclusion shall be improved, consider adding clinical applicability, for example, cost, accessibility, and patient readiness for VR interventions.

Author Response

We would like to thank the reviewer for his/her valuable comments, allowing us to improve our work and highlight the importance of this technological intervention in critically ill survivors.

Comments and Replies

  1. This is a systematic review manuscript, thus, it is crucial to comply to PRISMA guideline, details such as the exact search strings, date ranges, and any language restrictions should be included to enhance reproducibility and transparency of the systematic review process.

Reply: We have followed PRISMA guidelines, yet some details were not included. Thus, we have included timeline in the manuscript apart from the abstract and we also included search strategies as a table. Language restrictions are included in the methodology in the inclusion criteria.

  1. Include PRISMA checklist.

Reply: PRISMA checklist was provided to the editorial team after their request.

  1. Three distinct quality assessment tools (PEDro, NOS, and JBI) were included. However, the rationale for choosing different tools for different study designs should be clarified. 

Reply: In order to fully assess the methodological quality of the included studies and present possible bias, we decided to include more tools according to the design of its study. PEDro scale is best applied in randomized controlled trials, while NOS in non-RCT and JBI has been recognized as a tool for a series of cases. We believe that for each different study design, the most suitable tool should be chosen.

  1. The review includes only five studies with relatively small and heterogeneous samples (different populations, interventions, and outcomes). This limitation should be more explicitly acknowledged in the discussion, emphasizing how heterogeneity affects the generalizability of the conclusions regarding VR’s efficacy in post-ICU rehabilitation.

Reply: Indeed, due to the innovative nature of this intervention, we were able to identify very few studies. And when considering the diversity of Post-Intensive Care syndrome, heterogeneous outcomes were anticipated. This creates a significant limitation in generalizing the results, as the reviewer has noted. This was further emphasized at the discussion  (lines 314-317) “The small number of included studies and the heterogeneity of the outcomes presented do not allow the generalization of these preliminary findings. But present the various domains that may benefit from this therapeutic intervention”. And we further underlined this in the section of limitations that was already mentioned.

  1. Explain why meta-analysis is not performed/ included in the manuscript.

Meta-analysis was not performed due to the limited number of studies identified and the heterogeneity of the outcomes presented. Our goal was primarily to identify any studies implementing immersive VR after ICU discharge and to describe technological considerations that could be used either in further research or in clinical practice.

  1. The results indicate improvements in psychological, cognitive, and physical outcomes, however, the discussion does not fully differentiate the strength of evidence among these domains.

Although we were able to identify that all domains of post-intensive care syndrome were investigated, there was indeed differentiation to the extent that this occurred. Certainly, we agree with the reviewers’ point of view regarding the strength of evidence, as the limited number of published studies is a significant limitation. In order to make this more clear we changed our writing and we included the following (lines309-313) : “Positive findings were mostly noted in the domains of cognitive and psychological sequelae, like depression, anxiety, PTSD, and cognitive function. Physical impairments were not extensively examined. Yet, we were able to note a positive effect in functionality, but not in muscle strength.”

  1. The manuscript does not discuss the potential for publication bias or language bias, which could influence the synthesis since only English-language studies were included. Further explain/ justify this item.

Although the risk of bias wasn’t assessed through a specific tool as in meta-analysis, we included in the results the main risk of bias which was the bleeding processes. We incorporated the language restriction as most studies are published in English and due to the fact that translating studies from other languages could lead to falsification of data. To our knowledge, there are no relevant studies published in other languages. This could be presented as a limitation, and  we have included it in the specific section of the manuscript. We quote : “Additionally, language restriction could have limited the number of identified studies.”

  1. Conclusion shall be improved, consider adding clinical applicability, for example, cost, accessibility, and patient readiness for VR interventions.

We would like to thank the reviewer for underlining this. At the discussion we included (lines 393-397): “Patients noted an increased level of satisfaction, which indicates their ability to interact successfully in the virtual environment [14]. This is mostly attributed to newer software technology that allows a more accurate motion detection and interaction. This is an important feature when considering elderly populations who are not familiar with technology” and (lines 401-404) “This could have a significant limitation regarding the cost of a custom-made game, which leads to a significant access restriction. Whilst commercial games are easily accessible and with quite a low cost.”

Conclusions were rewritten: “Virtual reality technology could safely be implemented in the field of post-ICU recovery. It could be implemented for both educational and training purposes, providing a more enjoyable means of rehabilitation. Promising results were identified in enhancing the rehabilitation of physical, cognitive, and mental disorders of critically ill survivors.

The ability to provide dual task training and to combine both cognitive and physical components seems to augment training stimulus. Further research is needed to establish effectiveness and fully explore the sequelae of post-intensive care syndrome. Little evidence has been provided regarding functional rehabilitation and especially activities of daily living. Additionally, ICU population is quite heterogeneous, thus we need RCT studies with bigger populations that could fully describe all pathologies. As technology evolves further investigation of designing features, software and ways of enhancing interaction is of high importance, especially AI and decision supporting systems. Home-based training and monitoring of this population could also increase accessibility to such services.”

Reviewer 3 Report

Comments and Suggestions for Authors

A systematic review is a rigorous form of academic research that aims to collect, critically evaluate, and synthesize all available evidence on a specific research question using a structured, transparent, and reproducible methodology. 

In such a review, numerous studies should be systematically gathered, classified, and analyzed according to well-defined criteria such as methodology, dataset characteristics, performance metrics, or theoretical approaches. 

This process is intended to provide readers with an extended and comprehensive understanding of the research topic. 

However, in this paper, these essential focus areas appear to be missing. 

The analysis lacks sufficient depth and systematic categorization of existing studies, which significantly weakens the review’s overall contribution and its ability to present a clear overview of the current state of knowledge in the field.

MAIN OBJECTION

My main objection is by taking into account only 5 papers, it cannot be a systematic review.

Additionally, I have some major objections as follows:

1) The search query used in the paper is not clearly defined or adequately described. As a result, it is impossible to replicate or verify the search process and confirm whether all relevant studies were identified. A systematic review requires a transparent and reproducible search strategy; therefore, the lack of clarity in the query formulation significantly limits the reliability and reproducibility of the findings.

2) There are several inconsistencies and irregularities observed in the manuscript. 
For instance, the authors state that “The initial search of online databases identified 234 articles. After removing duplicates, 187 studies remained and were assessed for inclusion.” 
However, Figure 1 indicates that only 45 duplicates were removed, which does not align with the reported numbers in the text. 

This discrepancy raises concerns about the accuracy and transparency of the data screening process. 

Furthermore, certain parts of Figure 1 are poorly legible, making it difficult to interpret the flow of study selection and verify the consistency of the reported values. 

3)Some records appear to have been excluded in Figure 1 without any accompanying justification or explanation. In a systematic review, each stage of study exclusion should be clearly documented, with reasons provided for the removal of records to ensure transparency and reproducibility.

4)A systematic review is expected to include several comparative tables that summarize and contrast the key characteristics, methodologies, and findings of the studies included in the literature. 
Such tables provide a structured overview that allows readers to easily identify patterns, strengths, and gaps across different dimensions of the reviewed works. 
However, this paper presents only two tables, which is insufficient for capturing the full scope of the reviewed studies. 

5)A systematic literature review (SLR) should not only summarize existing research but also identify knowledge gaps and propose potential directions for future studies. 
Highlighting emerging trends, unresolved challenges, and underexplored topics is essential for guiding subsequent research efforts in the field. 
However, this paper does not provide any recommendations regarding future research directions. 

6)Limitations is an important section however it only contains 5-6 lines of text!

7) In academic papers, the Conclusion section holds significant importance, as it should summarize the main findings, emphasize their implications, and reflect on the overall contribution of the study. 
A well-written conclusion provides closure to the research and helps readers understand the broader relevance of the results. 
However, in this paper, the Conclusion section consists of only a single sentence, which is insufficient to convey the key outcomes or their significance. 

Author Response

Comment: A systematic review is a rigorous form of academic research that aims to collect, critically evaluate, and synthesize all available evidence on a specific research question using a structured, transparent, and reproducible methodology. 

In such a review, numerous studies should be systematically gathered, classified, and analyzed according to well-defined criteria such as methodology, dataset characteristics, performance metrics, or theoretical approaches. 

This process is intended to provide readers with an extended and comprehensive understanding of the research topic. 

However, in this paper, these essential focus areas appear to be missing. 

The analysis lacks sufficient depth and systematic categorization of existing studies, which significantly weakens the review’s overall contribution and its ability to present a clear overview of the current state of knowledge in the field.

Reply: We would like to thank the reviewer for looking into our manuscript and pointing out the importance of a systematic review. His/her comments are most valuable to further strengthen our manuscript. Indeed a systematic review is considered a research study that aims to gather and synthesize evidence on a specific research question. But also to summarize existing evidence. Our aim was specific to “the benefits across the spectrum of PICS, to describe the technological characteristics that could support and augment these interventions and present clinical recommendations. Thus our criteria and search strategies were designed according to it. We included not only RCTs in order to fully describe the technologies used apart from presenting the benefits in order to provide sufficient data on this topic, which is most innovative. And, we believe that the clinical considerations that we added provide an overview  and significant clinical knowledge.

MAIN OBJECTION

My main objection is by taking into account only 5 papers, it cannot be a systematic review.

Reply: We understand the concern raised by the reviewer, but to our knowledge there is no specific rule to state the number of studies to be included of a systematic review. This depends on the research question and the available evidence. Additionally, this is the results of the innovative nature of our research that focuses on a new technology applied in a quite unique population. Even by the limited number of studies, there are significant data to be gathered and most importantly this highlights the gaps in research of this filed. we have acknowledged and noted the limited number of included studies, as a limitation the limited number of included studies.

The limited number and heterogeneity of outcomes haven’t allowed a meta-analysis

Major objections as follows:

  • The search query used in the paper is not clearly defined or adequately described. As a result, it is impossible to replicate or verify the search process and confirm whether all relevant studies were identified. A systematic review requires a transparent and reproducible search strategy; therefore, the lack of clarity in the query formulation significantly limits the reliability and reproducibility of the findings.

Reply: Although we have included search terms according to PICO framework in order to make our search query clear, we understand that probably that wasn’t enough. Especially, when there is a need for replication of the search as the reviewer has noted. Thus, we included a table with the search strategies that were used specifically in pubmed, as this database allows a more complex search. Yet, if the reviewer believes that further information are need a supplemental file could be added. The extended number of databases that were searched, we believe, could ensure that we were able to identify all relevant studies.

2) There are several inconsistencies and irregularities observed in the manuscript. 
For instance, the authors state that “The initial search of online databases identified 234 articles. After removing duplicates, 187 studies remained and were assessed for inclusion.” 
However, Figure 1 indicates that only 45 duplicates were removed, which does not align with the reported numbers in the text. 

This discrepancy raises concerns about the accuracy and transparency of the data screening process. 

Furthermore, certain parts of Figure 1 are poorly legible, making it difficult to interpret the flow of study selection and verify the consistency of the reported values. 

Reply: We would like to thank the reviewer for bringing this to our attention. Probably, the transportation of the manuscript to the journals’ form changed our figure and led to the concealment of information. This was corrected and additional data were inserted.

And, thank you for pointing out a typing error concerning the “187 studies”  with the correct being  “189 studies”

3)Some records appear to have been excluded in Figure 1 without any accompanying justification or explanation. In a systematic review, each stage of study exclusion should be clearly documented, with reasons provided for the removal of records to ensure transparency and reproducibility.

Reply: We went over figure 1 and we included further information in order to ensure transparency.

4)A systematic review is expected to include several comparative tables that summarize and contrast the key characteristics, methodologies, and findings of the studies included in the literature. 
Such tables provide a structured overview that allows readers to easily identify patterns, strengths, and gaps across different dimensions of the reviewed works. 
However, this paper presents only two tables, which is insufficient for capturing the full scope of the reviewed studies. 

Reply: We believe that due to the limited number of studies, the tables included are adequate. Especially table 1 presents all relevant information regarding the population, the interventions and the outcomes used.  Thus, this allows the presentation of any differences among the studies. Additionally, as we wanted to further describe technological aspects we fully described the interventions in a separate paragraph. Yet, if the reviewer believes that our manuscript would be benefited by adding further tables, we could add them.

5)A systematic literature review (SLR) should not only summarize existing research but also identify knowledge gaps and propose potential directions for future studies. 
Highlighting emerging trends, unresolved challenges, and underexplored topics is essential for guiding subsequent research efforts in the field. 
However, this paper does not provide any recommendations regarding future research directions. 

Reply: We would like to thank the reviewer for bringing this to our attention. Although we summarized a few recommendation, these could be further presented. Thus, we included the following: “ Further research is needed to establish effectiveness and fully explore the sequelae of post-intensive care syndrome. Little evidence has been provided regarding functional rehabilitation and especially activities of daily living. Additionally, ICU population is quite heterogeneous, thus we need RCT studies with bigger populations that could fully describe all pathologies. As technology evolves further investigation of designing fea-tures, software and ways of enhancing interaction is of high importance, especially AI and decision supporting systems. Home-based training and monitoring of this popula-tion could also increase accessibility to such services”

6)Limitations is an important section however it only contains 5-6 lines of text!

Reply: We further enriched the limitations section as follows: “At this point, only a limited number of randomized controlled trials are available concerning post-ICU rehabilitation. The sample size of the included population is often limited and not representative of the ICU population. The protocols that were implemented and the outcome measures that were assessed are very heterogeneous. Physical functioning and activities of daily living weren’t fully explored. And, other symptoms related to post-intensive care syndrome, like pain, sleep disorders, endurance etc. Thus, definitive conclusions cannot be drawn, and these results cannot be generalized. Additionally, language restrictions could have limited the number of identified studies, especially from countries that have shown significant technological evolution, such as China or Japan. Yet, this is a promising field that should be explored further.” Lines 445-455.

7) In academic papers, the Conclusion section holds significant importance, as it should summarize the main findings, emphasize their implications, and reflect on the overall contribution of the study. 
A well-written conclusion provides closure to the research and helps readers understand the broader relevance of the results. 
However, in this paper, the Conclusion section consists of only a single sentence, which is insufficient to convey the key outcomes or their significance. 

Reply: Conclusion were written and we included further recommendations on future research as proposed. Lines (458-470) “Virtual reality technology could safely be implemented in the field of post-ICU recovery. It could be implemented for both educational and training purposes, providing a more enjoyable means of rehabilitation. Promising results were identified in enhancing the rehabilitation of physical, cognitive, and mental disorders of critically ill survivors.

The ability to provide dual task training and to combine both cognitive and physical components seems to augment training stimulus. Further research is needed to establish effectiveness and fully explore the sequelae of post-intensive care syndrome. Little evidence has been provided regarding functional rehabilitation and especially activities of daily living. Additionally, ICU population is quite heterogeneous, thus we need RCT studies with bigger populations that could fully describe all pathologies. As technology evolves further investigation of designing features, software and ways of enhancing interaction even in more frail and weak patients is of high importance. Home-based training and monitoring of this population could also increase accessibility to such services.”

Reviewer 4 Report

Comments and Suggestions for Authors

Dear authors,

I have now completed the review of the manuscript titled Could immersive virtual reality facilitate recovery of survivors of critical illness? A systematic review.

The manuscript is interesting and, in general, fairly well-written.

However, I still have some suggestions to further improve the quality of the manuscript.

I would like to suggest that the authors address these limitations in the article, either by discussing them in the limitations section or, where feasible, by making the appropriate revisions:

1. The heterogeneity across included studies is problematic. The interventions varied substantially, from 14-minute informational videos to multi-session cognitive training programs. Similarly, the timing of interventions ranged from a few days post-ICU discharge to several months after hospital discharge. This variation makes meaningful comparison across studies virtually impossible and undermines the validity of synthesizing results.

2. Some recent findings could be stated in the introduction. For example, Explainable AI in Clinical Decision Support Systems: A Meta-Analysis of Methods, Applications, and Usability Challenges - This would help readers understand how AI technologies (including VR systems) are being integrated into clinical settings and the challenges around usability, which directly relates to implementing VR interventions for ICU survivors.

3. The quality assessment reveals concerning gaps. While the authors report generally good methodological quality using the PEDro scale (mean score 5.6 out of 10), this represents only moderate quality at best. One study scored as low as 5 out of 10, indicating fair quality. The inclusion of a case series study further weakens the overall evidence quality, as such designs lack the control necessary to establish causation.

4. Discussion would be extended by briefly mentioning latest research, to show readers future research possibilities. For example, Exploring the Role of Artificial Intelligence in Smart Healthcare: A Capability and Function-Oriented Review - This provides broader context for how emerging technologies like VR fit within the smart healthcare ecosystem, which would deepen readers' understanding of VR's role in rehabilitation.

5. The review is purely narrative and lacks meta-analysis. While the heterogeneity may justify this decision, the authors provide no quantitative synthesis of effect sizes across outcomes. Without pooled estimates or forest plots, readers cannot assess the magnitude or consistency of treatment effects. The review essentially presents a descriptive summary rather than a rigorous quantitative synthesis.

Thank you for your valuable contributions to our field of research. I look forward to receiving the revised manuscript.

Author Response

We would like to thank the reviewer for his/her positive comments that would allow us to further enhance our manuscript.

Comments

  1. The heterogeneity across included studies is problematic. The interventions varied substantially, from 14-minute informational videos to multi-session cognitive training programs. Similarly, the timing of interventions ranged from a few days post-ICU discharge to several months after hospital discharge. This variation makes meaningful comparison across studies virtually impossible and undermines the validity of synthesizing results.

Reply: Due to the novelty of this intervention, especially in critically ill patients, we expected to have limited numbers of studies. Additionally, the multidimensional nature of post-intensive care syndrome could be also the main cause of the heterogeneity presented in both the interventions and the outcomes selected. Taking into consideration that even in traditional rehabilitation services, the optimal time for referral of the survivor isn’t known, it is anticipated this to be noted here as well. These are noted as limitations and we ameliorated our  findings in the discussion and conclusions.

  1. Some recent findings could be stated in the introduction. For example, Explainable AI in Clinical Decision Support Systems: A Meta-Analysis of Methods, Applications, and Usability Challenges - This would help readers understand how AI technologies (including VR systems) are being integrated into clinical settings and the challenges around usability, which directly relates to implementing VR interventions for ICU survivors.

We would like to thank the reviewer for bringing this to our attention, as it provides further insight into the technological features of VR systems. We believe that it is best to add this in the section where we discuss these technological points, our clinical recommendations. (lines 431-432 and 436-441).

  1. The quality assessment reveals concerning gaps. While the authors report generally good methodological quality using the PEDro scale (mean score 5.6 out of 10), this represents only moderate quality at best. One study scored as low as 5 out of 10, indicating fair quality. The inclusion of a case series study further weakens the overall evidence quality, as such designs lack the control necessary to establish causation.

Reply: Indeed, we agree with the reviewer's comment and it is not our attention to overestimate the methodological quality of the included studies. In order to be more precise as possible we included different tools in accordance to the design of its study. In order to fairly present this we included the following: “In general, we noted a moderate methodological quality. Blinding was absent in all studies.” as a general comment.

  1. Discussion would be extended by briefly mentioning latest research, to show readers future research possibilities. For example, Exploring the Role of Artificial Intelligence in Smart Healthcare: A Capability and Function-Oriented Review - This provides broader context for how emerging technologies like VR fit within the smart healthcare ecosystem, which would deepen readers' understanding of VR's role in rehabilitation.

Indeed, AI has been used to support and advance VR rehabilitation. We included the references provided by the reviewer to further explain technological features of our clinical recommendations. (lines 431-432 and 436-441)

Additionally, we included this in our recommendation on future research.

  1. The review is purely narrative and lacks meta-analysis. While the heterogeneity may justify this decision, the authors provide no quantitative synthesis of effect sizes across outcomes. Without pooled estimates or forest plots, readers cannot assess the magnitude or consistency of treatment effects. The review essentially presents a descriptive summary rather than a rigorous quantitative synthesis.

Reply: When considering the limited number of included studies, it is difficult to have a meta-analysis and a quantitative synthesis with such heterogeneous outcomes. Yet, we believe that we made a significant effort to present a qualitative synthesis of the results, while underlining the limitations of this. As our aim was to present any data from implementing VR in critically ill survivors and to present clinical recommendations, we believe that these have been achieved.  Additionally, we believe that looking into designing features and describing the scenarios has a value for future research and clinical implementation of this technology. A meta-analysis and a quantitative synthesis would be allowed in the future after the publication of more studies.

Thank you for your valuable contributions to our field of research. I look forward to receiving the revised manuscript.

We appreciate all the comments made and reflections that were presented and shared.

Round 2

Reviewer 1 Report

Comments and Suggestions for Authors

I appreciate the authors for their time and efforts to revise the manuscript. However, risk of bias assessment and providing a synthesis is part of the systematic review reporting.

Author Response

Comment

I appreciate the authors for their time and efforts to revise the manuscript. However, risk of bias assessment and providing a synthesis is part of the systematic review reporting.

Reply.

We would like to thank the reviewer for his/her valuable comment.

The diversity in the design of the included studies led us to use three different tools to assess the methodological quality of the included studies. These tools offer as well information on risk of bias. As the reviewer has noted the need to provide further information we added the following: “In general, we noted a moderate methodological quality. A major concern could be identified in blinding of subjects, assessors and therapists that was absent in all studies. Additionally, the absence of intention-to-treat analysis could be regarded as a risk of bias, as its application provides a more accurate picture of the interventions’ effectiveness.” Lines 177-181. We also added a description for the number of stars that the study of Despoti et al gathered as assessed by NEWCASTLE - OTTAWA QUALITY ASSESSMENT SCALE (line 185-187).

If the reviewer finds that these aren’t adequate, we could still use the RoB 2 tool risk of bias. Yet, we feel that as only three studies are RCTS, this wouldn’t be of much value. But still, if the reviewer needs it to be added, we could do so.

Reviewer 2 Report

Comments and Suggestions for Authors

The authors have addressed the comments accordingly.

Author Response

Comment

The authors have addressed the comments accordingly.

Reply.

We would like to thank the reviewer for all the valuable comments.

Reviewer 3 Report

Comments and Suggestions for Authors

This is my second-round evaluation of the manuscript titled “Could immersive virtual reality facilitate recovery of survivors of critical illness? A systematic review.” 

The article is clearly identified as a Systematic Review; however, the scope and methodology do not meet the standards typically expected of such papers.

Upon examination, it appears that the authors have only reviewed and compared five studies. 

This is an extremely limited dataset for a paper presented as a systematic review. 

Even in standard conference or short research papers, it is common to include a broader range of literature for comparison and synthesis.

While the topic itself is relevant and potentially valuable, the current form of the manuscript does not align with the depth and methodological standards expected for a systematic review. 

Therefore, I maintain a strong objection to the acceptance of the paper in its current form.

Author Response

Comment

This is my second-round evaluation of the manuscript titled “Could immersive virtual reality facilitate recovery of survivors of critical illness? A systematic review.” 

The article is clearly identified as a Systematic Review; however, the scope and methodology do not meet the standards typically expected of such papers.

Upon examination, it appears that the authors have only reviewed and compared five studies. 

This is an extremely limited dataset for a paper presented as a systematic review. 

Even in standard conference or short research papers, it is common to include a broader range of literature for comparison and synthesis.

While the topic itself is relevant and potentially valuable, the current form of the manuscript does not align with the depth and methodological standards expected for a systematic review. 

Therefore, I maintain a strong objection to the acceptance of the paper in its current form.

Reply

We would like to thank the reviewer for pointing out his/her concerns. Virtual reality in the field of ICU rehabilitation is an innovative field and has been introduced in research during the past few years. But, after discharge and regarding post-ICU rehabilitation regarding the post-intensive care syndrome, little has been explored. As, the research and clinical community has been raising awareness on PICS, we believe that it is of high importance to present this technology as another option to manage PICS. Our research team has been involved  in critically ill patients' rehabilitation and the use of technology, and this probably also explains our interest on this topic. In order to underline the importance of this field, I would like to add that just a few days ago the European Intensive Care Society just announced another scientific section that of PICS. We strongly feel that the novelty of a subject often leads to a limited number of studies being included in a systematic review, and this is not uncommon as seen in other systematic reviews as well and explains the lack of a specific guideline on the number of included studies. 

Reviewer 4 Report

Comments and Suggestions for Authors

All comments addressed.

Author Response

Comment.

All comments addressed.

Reply.

We would like to thank the reviewer for his/her valuable comments.